# Estimating Forest Stand Height in Savannakhet, Lao PDR Using InSAR and Backscatter Methods with L-Band SAR Data

**Helen Blue Parache** [1,2,*]**, Timothy Mayer** [1,2]**, Kelsey E. Herndon** [1,2]**, Africa Ixmucane Flores-Anderson** [1,2]**, Yang Lei** [3]**, Quyen Nguyen** [4]**, Thannarot Kunlamai** [5] **and Robert Griffin** [1,2]

1   Earth System Science Center, The University of Alabama Huntsville, 320 Sparkman Drive, Huntsville, AL 35805, USA; timothy.j.mayer@nasa.gov (T.M.); kelsey.e.herndon@nasa.gov (K.E.H.); africa.flores@nasa.gov (A.I.F.-A.); reg0005@uah.edu (R.G.)
2   SERVIR Science Coordination Office, NASA Marshall Space Flight Center, 320 Sparkman Drive, Huntsville, AL 35805, USA
3   Division of Geological and Planetary Science, California Institute of Technology, Pasadena, CA 91125, USA; ylei@caltech.edu
4   SERVIR-Mekong, Asian Disaster Preparedness Center, Bangkok 10400, Thailand; nguyen.quyen@adpc.net
5   Thaicom Public Company Limited, Bangkok 10400, Thailand; thannarot.kunlamai@adpc.net
*   Correspondence: helen.b.baldwin@nasa.gov; Tel.: +1-256-961-7002

**Abstract:** Forest stand height (FSH), or average canopy height, serves as an important indicator for forest monitoring. The information provided about above-ground biomass for greenhouse gas emissions reporting and estimating carbon storage is relevant for reporting for Reducing Emissions from Deforestation and Forest Degradation (REDD+). A novel forest height estimation method utilizing a fusion of backscatter and Interferometric Synthetic Aperture Radar (InSAR) data from JAXA's Advanced Land Observing Satellite Phased Array type L-band Synthetic Aperture Radar (ALOS PALSAR) is applied to a use case in Savannakhet, Lao. Compared with LiDAR, the estimated height from the fusion method had an RMSE of 4.90 m and an $R^2$ of 0.26. These results are comparable to previous studies using SAR estimation techniques. Despite limitations of data quality and quantity, the Savannakhet, Lao use case demonstrates the applicability of these techniques utilizing L-band SAR data for estimating FSH in tropical forests and can be used as a springboard for use of L-band data from the future NASA-ISRO SAR (NISAR) mission.

**Keywords:** interferometry; backscatter; MRV; REDD+; ALOS PALSAR; SAR; forest monitoring; Lao





## 1. Introduction

### 1.1. REDD+ in Lao PDR

The Lao People's Democratic Republic (PDR) committed to the REDD+ program (Reducing Emissions from Deforestation and Forest Degradation plus the sustainable management of forests, and the conservation and enhancement of forest carbon stocks), in 2007, by joining the Forest Carbon Partnership Facility [1]. A United Nations Framework Convention on Climate Change (UNFCCC) program, REDD+ aims to provide incentives for the reduction of emissions caused by deforestation and forest degradation [2]. Through these incentives for participating member countries, the REDD+ program aims to protect and enhance the capacity of forests to act as natural carbon sinks [3]. The Forest Reference Emission Level (FREL) is an integral component of REDD+, providing a baseline for evaluating the impact of REDD+ activities through changes in emissions [4].

Since 2007, the institutional landscape implementing REDD+ in Lao PDR has evolved. The Ministry of Agriculture and Forest established the national REDD+ Task Force in 2008 [5], which has since been reorganized into the Department of Forestry (DoF) and Ministry of Agriculture and Forestry (MAF) [1]. The Ministry of Natural Resources and Environment (MoNRE) was established in 2011. Currently, the REDD+ Division in the MoNRE and the REDD+ Office in the MAF are both responsible for REDD+ activities, and

occupy the same level in the administrative hierarchy [6]. The first Readiness Grant was signed in 2014, for USD 3.6 million. Lao PDR's Emission Reduction Program Document (ER-PD) was accepted in June 2018 at the 18th Carbon Fund Participants Meeting and proposed to reduce emissions in six provinces by targeting small-scale drivers of forest degradation, e.g., shifting cultivation [1]. REDD+ action plans have been made for these six REDD+ pilot provinces [6].

Traditionally, the forest monitoring required for these reports has been accomplished through field work, but this is challenging due to the cost and time investment required [7], especially for a country-level initiative. As a result, the most recent guidance includes recommendations incorporating remote sensing as a mechanism for broad-scale and sustained monitoring efforts [8,9]. Currently, medium resolution optical data are often used. However, optical data are considered more useful for detecting land cover changes, such as deforestation, while radar is more suitable for estimating above-ground biomass (AGB) [10]. Lao PDR used remote sensing techniques for mapping land cover types and change detection along with country-specific allometric equations for the proposed FREL [11]. In 2018, the UNFCCC secretariat and technical assessment team reviewed Lao PDR's proposed FREL and recommended that Lao PDR enhance capacity in remote sensing for forest mapping and estimation of forest degradation and subsequent emissions [11]. This article aims to support those recommendations through investigating a use case of novel L-band SAR methods for estimating forest stand height (FSH), or mean canopy height, in Savannakhet, Lao PDR.

*1.2. Estimating Forest Stand Height with Remote Sensing*

AGB is often used as a proxy measurement of carbon storage or sequestration [7]. The Intergovernmental Panel on Climate Change (IPCC) suggests two approaches, Biomass Gain-loss and Stock-Difference, for estimating carbon storage depending on data availability and technical capacity. AGB is essential for estimating overall biomass for both approaches [12]. FSH is a key characteristic of vegetation structure that can be used within allometric equations to estimate AGB [13,14].

Remote sensing is beneficial for forest monitoring as it can cover large areas and provide regular updates on forest cover and quality [14]. Optical sensors, such as Landsat 7 and 8, have been used for estimating FSH with some success [7,15–17]. Airborne LiDAR (Light Detection and Ranging) is often used alongside optical data, providing more precise vertical data to train models using optical data as inputs to estimate forest height [18]. Multiple regression and Random Forest approaches utilizing Landsat and LiDAR were evaluated for estimating FSH in a temperate forest in British Columbia, Canada, with $R^2$ of 0.61 and root mean square error (RMSE) of 4.18 m and $R^2$ of 0.82 and RMSE 3.17 m respectively, at a minimum object size of 2.0 ha [19]. Terrestrial LiDAR has also been used to estimate structure parameters of forests for estimating biomass with high accuracy [20]. Recently, the improved coverage of spaceborne laser data through the Global Ecosystem Dynamics Investigation (GEDI) mission has allowed for more comprehensive training of optical datasets, such as Landsat, to estimate FSH, resulting in a global RMSE of 7.08 m predicted using a Random Forest model at 3 km resolution [21]. Another recent study utilized Landsat and GEDI to create a global forest height map at 30 m resolution with an $R^2$ of 0.61 and RMSE of 9.07 m compared with LiDAR [22].

SAR offers potential as compared to optical approaches, as SAR measures vertical structure as opposed to the 'greenness' measured by optical sensors [15], tending to saturate at low AGB levels [23]. In addition, SAR is an active remote sensing system and thus able to acquire usable data despite cloud cover [24], which is especially important for tropical regions such as Lao PDR [25]. A variety of methods based on Synthetic Aperture Radar (SAR) have been developed to take advantage of SAR's sensitivity to structure [7,26,27]. Multiple linear regressions at 100 m spacing have been utilized to compare the performance of optical and SAR datasets, resulting in an $R^2$ of 0.15 for European Remote-Sensing Satellite

1/2 C-band coherence data, 0.08 $R^2$ for Japanese Earth Resources Satellite 1 L-band data, and 0.26 for Landsat data [28].

Full polarimetric SAR has also been used to derive FSH in a tropical forest and provides results using machine learning (ML) methods [29], integrating LiDAR data and multiple polarimetric variables extracted using quad-pol decomposition techniques to achieve an average $R^2$ equal to 0.7 and RMSE equal to 10 m. This study shows that full polarimetric data and LiDAR can be used to derive FSH. However, the overall RMSE values are high and, as expected, the performance of the ML algorithms were highly dependent on the training data. InSAR-based Random Volume over Ground (RVoG) methods were developed and enhanced to allow for repeat pass data to be utilized [30]. These techniques have been continued to be for FSH estimation [31–34]. The method utilized in this study incorporates the effect of the dielectric change on the target into models previously developed by [31] and [35]. The InSAR method utilized in this study was developed for situations when data availability did not allow for the applications of other PolInSAR techniques [33].

Backscatter power has been used as a basis for forest height estimation [36,37] and for improving estimates that also incorporate optical data [38]. As backscatter tends to saturate at high AGB levels, the fusion of backscatter and InSAR-based methods was shown to be valuable in a previous study [39].

InSAR techniques have not been extensively applied to tropical regions, although a few recent examples exist [40,41]. One study achieved an $R^2$ equal to 0.54 to 0.69 and an RMSE of 4.65 to 8.44 m at 3 to 12 m over tropical forest [40]. However, it is essential to prioritize the investigation of methods taking advantage of data that are openly accessible when considering the needs of end users. Studying the effectiveness of SAR-based methods in a variety of ecoregions will enable decision makers in countries participating in REDD+ to more confidently implement FSH estimation methodologies that rely on SAR data. This is especially important in tropical forests, which store up to 3.5 times more carbon than other forest types and make up over half of the total global terrestrial carbon storage [42,43].

### 1.3. Area of Interest

Savannakhet Province, located in the southern part of Lao PDR, contains three national biodiversity conservation areas: Phou Xang He (109,900 hectares), Dong Phou Vieng (197,000 hectares), and Xe Bang Noun (150,000 hectares) [44]. Clearing land for hydropower projects, mining, and other economic activities is the main driver of deforestation and forest degradation in Lao PDR [1]. As of 2000, rich forests still comprised 70 percent of the province's total 21,774 square kilometer area [45]. These forests have subsequently experienced increased fragmentation and deforestation. As of 2010, forest cover only accounted for 41 percent of the province [46,47]. The study area in Savannakhet, Lao PDR is comprised of forest, surface water, orchard or plantations, evergreen broad leaf, mixed forest, urban and built up, cropland, barren, wetlands, grassland, shrubland, and aquaculture as defined by the SERVIR-Mekong Regional Land Cover Monitoring System (RLCMS) [48].

This region experiences a tropical monsoon climate, with an average annual rainfall of 1440 mm [16]. The wet season extends from May to October, and the dry season from November to April [44]. However, the rainfall is much greater in the eastern region than the western [45]. To the west, Savannahket's lowlands are bounded by the Mekong River. To the east, Savannakhet includes the Annamite Range [45]. The Dongsithouane Production Forest (PF) (Figure 1) is located in the southwest corner of Savannakhet Province at 16°33′N and 104°45′E and is surrounded by the Banghiang River on three sides. The Dongsithuane PF contains mostly deciduous tree species, primarily from the Dipterocarpaceae family, with canopy cover ranging from 10 to 70 percent [44]. The median height is 9 m and the maximum height is 79 m. The map and basic statistics of the LiDAR-based FSH can be found in Figure 3a. The elevation in this forest ranges from 95 to 265 m [44]. This PF was part of a successful pilot community forestry project in the 1990s [47]. The 2007 Forestry Law describes PFs as any forest used for socioeconomic purposes [49].

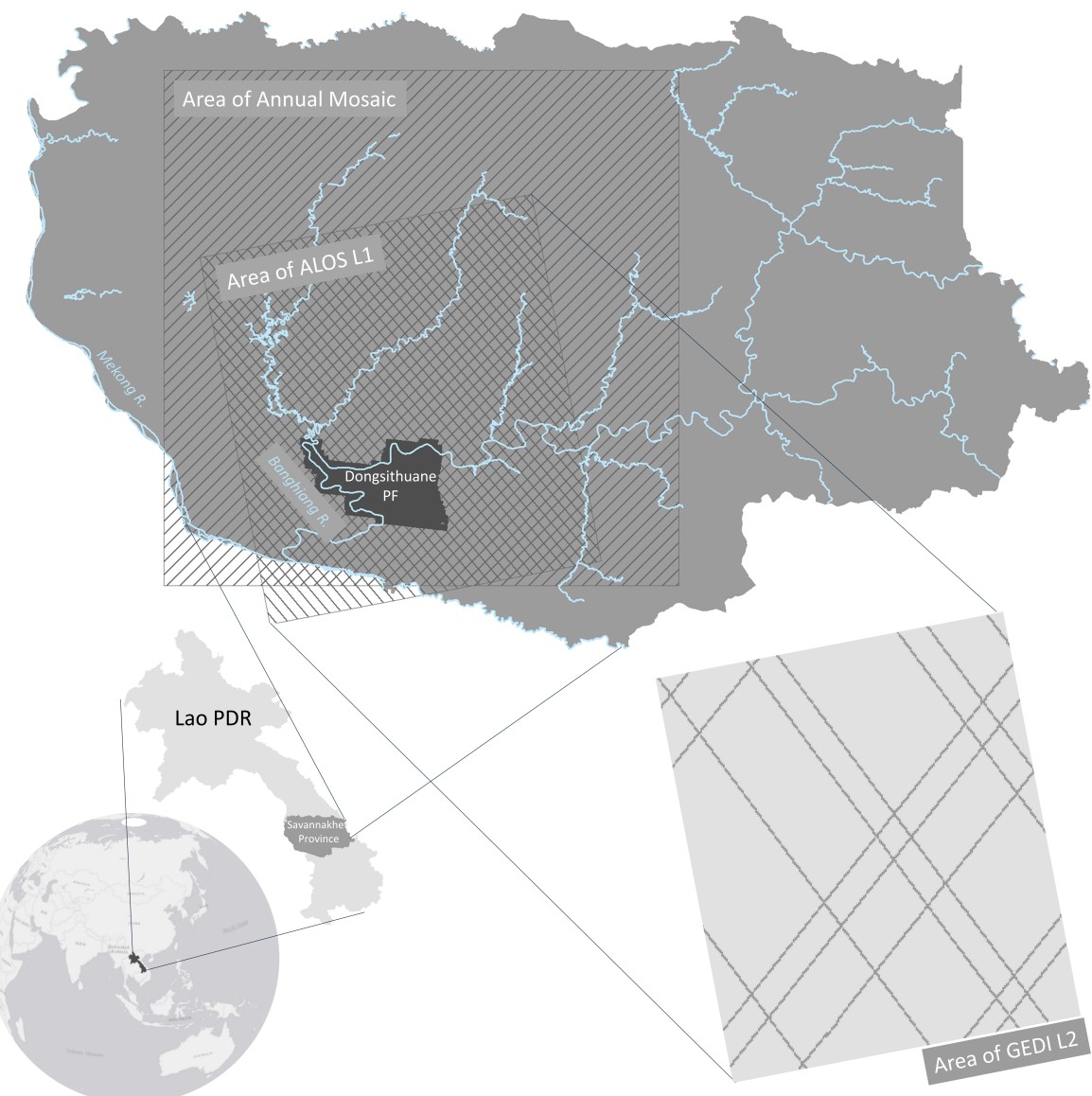

**Figure 1.** LiDAR vegetation height, used as a proxy for ground truth FSH, was available for the Dongsithouane PF located in Savannakhet Province, Lao PDR. The Dongsithouane PF is surrounded on three sides by the Banghiang River [50]. JAXA's ALOS L1 and the annual mosaic products encompass the area covered by LiDAR data. Height estimates were compared over the crosshatched area. The RLCMS and GLAD datasets were available for the entire crosshatched area. The GEDI L2 availability over the fusion height estimate area can be seen in dark grey in the bottom right.

### 1.4. Objectives

This study examines the application of an FSH estimation method that combines backscatter and interferometric SAR (InSAR) techniques to estimate FSH in a tropical forest in Lao PDR. Previously used to estimate FSH for the state of Maine, U.S. [33,39,51], this use case demonstrates the applicability of the FSH estimation method in a tropical region with limited data availability. Basic performance statistics, $R^2$, RMSE, and bias were used to assess the performance of the backscatter, InSAR, and fusion FSH estimates. This study provides insight into data acquisition challenges and any advantages in performance afforded by including InSAR. In addition, this study compares the height estimates with GEDI L2 Footprint Level Canopy Height, version 1 [52] and the Global Land Analysis and Discovery (GLAD) Global Forest Canopy Height for 2019 [22].

## 2. Materials and Methods

### 2.1. Data

This study relied on L-band SAR data and LiDAR data to estimate FSH. Datasets from GEDI and GLAD were used to compare with the estimated forest height output maps. Ancillary datasets provided essential information about land cover and elevation. These datasets are summarized with their respective native spatial for the specific scene investigated and temporal resolutions in Table 1.

**Table 1.** Summary of datasets.

| Dataset | Native Spatial Resolution | Temporal Resolution | Dates Used |
|---------|---------------------------|---------------------|------------|
| ALOS L1 interferograms | 30 m | 46 days | 13 June and 29 July 2009 |
| Annual mosaic | 24 m | annual | 13 June, 30 September and 12 October 2009 |
| LiDAR | 30 m | - | 6–8 February 2009 |
| RLCMS | 30 m | annual | 2009 |
| GLAD 2019 | 30 m | - | 2019 |
| GEDI L2 | 25 m diameter | - | 2019–2020 |
| SRTM | 25 m | - | 2000 |

### 2.1.1. ALOS PALSAR

The Japan Aerospace Exploration Agency (JAXA)'s Advanced Land Observation Satellite Phased Array type L-Band Synthetic Aperture Radar (ALOS PALSAR) instrument, launched in 2006, provides the only currently free and open L-band SAR data available [26]. Long wavelength radar, such as L-band, is sensitive to AGB density. The lower frequency of the approximately 24 cm wavelength decreases attenuation in the canopy, allowing it to gather information about the larger limbs and tree trunks [27,36]. ALOS-1 has a return period of 46 days [53].

Six scenes of ALOS PALSAR L-Band SAR level 1.0 raw/unprocessed (ALOS L1) imagery were downloaded from the Alaska Satellite Facility (ASF) for path 477 orbit 310 as an input for the InSAR technique [54]. The data were collected in fine mode, returning horizontal transmit horizontal receive (HH) and horizontal transmit vertical receive (HV) polarizations with a range resolution of 14 to 88 m [53]. The HV polarization was selected for this study since it is more sensitive than HH to vegetation structure [26,36]. Cross polarized measurements, such as HV, are the most appropriate choice for measuring vegetation, since their signal is dominated by the canopy (i.e., volumetric scattering) rather than the surface. This is because the contrast between H and V components is less for vegetation than for bare ground [26,27]. As an input to the InSAR technique, HH polarization creates a relationship between actual and estimated heights that is less linear than the HV polarization, causing more error in low and high height estimates [33].

The annual mosaic for 2009 was acquired from JAXA. These data provide ortho and slope corrected backscattering coefficients [55]. The annual mosaic includes data from 13 June, 30 September, and 12 October 2009. The annual mosaic was used as an input for the backscatter-based height estimation technique. The tile N17E105_09 covers the study area at a 25 m resolution.

### 2.1.2. Training and Testing Datasets

The proxy for ground truth FSH dataset used for training and testing both backscatter and InSAR models was Airborne Laser Scanning (ALS)-based height data acquired 6–8 February 2009 [23,44]. These LiDAR data were compared with vegetation heights acquired through field data collection conducted for the Dongsithouane Production Forest in February 2009, with an $R^2$ of 0.91. The data were acquired with a sampling density of 1 pulse/m$^2$ by Finnmap and Arbonaut [23] from a Piper PA-31 Navajo aircraft, using a Leica ALS 40 LiDAR scanner with a field of view of 30 degrees. The first and last echoes were used to create a height map of 30 m resolution over the 25,000 ha Dongsithuane Production Forest in Savannakhet, Lao PDR [44].

In addition to the LiDAR testing dataset, two other datasets were used as testing datasets, i.e., to compare with the backscatter, InSAR, and fusion technique-based height estimates in this location. GLAD's Global Canopy Cover dataset for 2019 [22] was used to compare height estimates over the entire area of the fusion estimate. This dataset was produced in Google Earth Engine (GEE) based on data from the Landsat series and incorporating GEDI Level 2 data for calibration and validation [22]. Originally a 30 m dataset, the GLAD product was resampled to 245 m for this comparison. In addition, GEDI, space-based LiDAR, was used to compare with the three FSH estimates produced for this study. The GEDI L2 Footprint Level Canopy Height product was selected. The gridded L3 product is a simple average of all points within 1 km [56] although a more complex method is being planned, and so utilizing the L2 product was expected to provide greater accuracy than the gridded L3 product. The footprint of each LiDAR point in the L2 product is 25 m in diameter [52].

### 2.1.3. Regional Land Cover Monitoring System

Non-forested areas were masked out of the study area in order to improve the accuracy of the InSAR method. For example, urban or agricultural areas may introduce large amounts of phase change between scenes, influencing the model that is created during FSH estimation [26]. In order to determine which areas of the study area should have the FSH estimation method applied, the Regional Land Cover Monitoring System (RLCMS) for the Mekong region [48] was used to identify land cover. These maps have been produced annually since 1987 to the present. The year 2009 was selected for this project to match most closely with the LiDAR, proxy ground truth, data.

The RLCMS approach utilizes 18 land cover classes [48]. The study area in Savannakhet, Lao PDR is comprised of 12 land cover classes. Forest, orchard or plantation forest, evergreen broadleaf, mixed forest and barren land cover classes were included as forest in the forest/non-forest mask. Including barren land improved the dynamic range, providing height values of 0. Surface water, urban and built up, cropland, wetlands, grassland, shrubland, and aquaculture were included as non-forest.

### 2.1.4. Shuttle Radar Topography Mission

In order to use the InSAR technique, date pairs of ALOS L1 data must be processed into one interferogram. This is achieved using the NASA Jet Propulsion Laboratory's Interferometric SAR Scientific Computing Environment (ISCE) v2.4.1 stripmapApp package [57,58]. During this process, the appropriate Shuttle Radar Topography Mission (SRTM) Digital Elevation Model (DEM) [59] was automatically downloaded and applied from NASA's Land Processes Distributed Active Archive Center (LP DAAC) [57] to aid in the geocoding and coregistration processes. In addition, the DEM was used to remove the topographic InSAR phase from the apparent interferogram.

### 2.1.5. CHIRPS

Climate Hazards Group InfraRed Precipitation with Station data were used to assess the potential impact of weather on dielectric properties across the scene [60]. These data were acquired at 0.05-degree resolution for the three days up to and including the day of the sensor observation. See Appendix A for this investigation.

### 2.2. Methods

This study investigates the fusion of two SAR-based approaches for estimating FSH (Figure 2). While the backscatter technique has been used to estimate vegetation height, it has been shown to saturate after about 10 m in height [51]. The InSAR technique improved some validation statistics when these estimates have been used in tandem with the backscatter estimates, replacing backscatter estimates above a 10 m threshold [26,33].

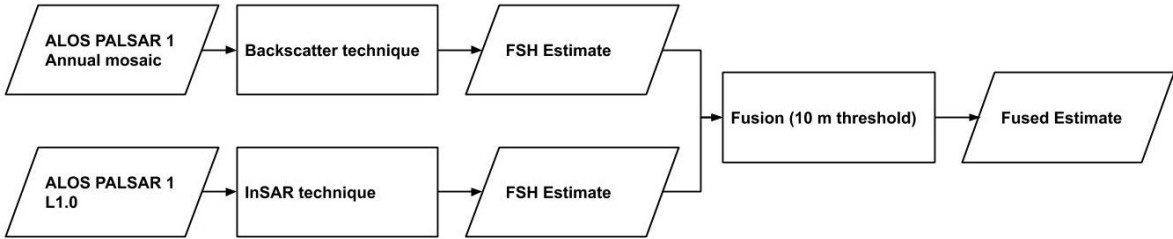

**Figure 2.** Simplified workflow for estimating forest stand height with ALOS PALSAR L-band data using backscatter and InSAR methods over Savannakhet, Lao PDR.

### 2.2.1. Backscatter Technique

The backscatter coefficient, $\sigma_0$, measured by the ALOS PALSAR sensor is the proportion of power returned to the sensor, as defined in Equation (1). In this equation, $I_{received}$ is the intensity of power received back at the sensor and $I_{incident}$ is the intensity of incident power [27].

$$\sigma_0 = \frac{I_{received}}{I_{incident}} 4\pi R^2 \tag{1}$$

Backscatter power has a direct relationship with vegetation height [36,51]. As the height of the vegetation increases, the more scatterers there are, and the greater the backscatter power received by the sensor [27]. The backscatter technique (Figure 3) for estimating FSH relies on the assumption that the scattering intensity in a forest stand increases with vegetation height. However, backscatter loses sensitivity to tree height with taller forest stands [51], as only the upper canopy returns backscatter in forests with high biomass [27,36]. This saturation issue can be addressed by incorporating the InSAR method [33] (see Section 2.2.3).

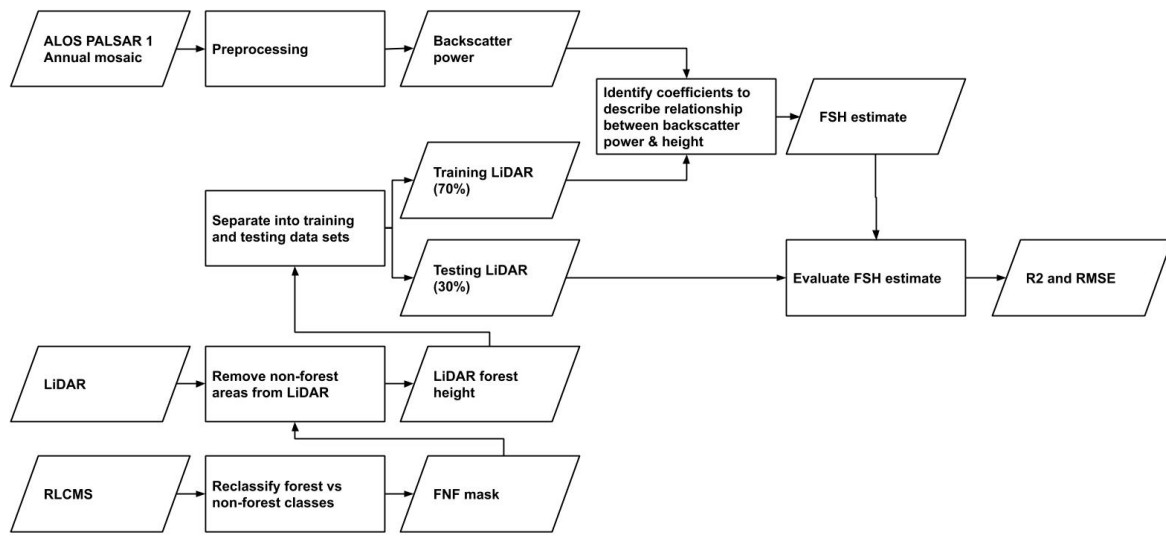

**Figure 3.** Workflow for estimating forest stand height with ALOS PALSAR L-band data using a backscatter-based method.

The input for the backscatter estimation employed the annual mosaic for 2009 and the ground truth proxy LiDAR vegetation height. JAXA radiometrically and geometrically corrects the annual mosaic [61]. Previous studies have successfully used the JAXA annual mosaic for estimating vegetation characteristics from backscatter power [33,36]. ISCE v2.4.1 does not support radiometric terrain correction for the ALOS PALSAR sensor, and so ALOS L1 data could not be used. The ASF Radiometric Terrain Correction (RTC) product for ALOS PALSAR was investigated in addition to the annual mosaic (Appendix C) but ultimately was not selected for this analysis as it did not result in a significant improvement

in results. The JAXA annual mosaic product must be converted from digital numbers (DN) representing amplitude into $\gamma_0$ in decibel values $\gamma_{db}$ using Equation (2) [26]:

$$\gamma_{db} = 10 \log_{10}[(DN)^2] - 83.0. \tag{2}$$

Then, $\gamma_{db}$ was transformed into $\gamma_{pw}$, or backscatter power, using Equation (3) [26]:

$$\gamma_{pw} = 10^{0.1\gamma_{db}}. \tag{3}$$

The backscatter datasets were then resampled from 25 to 245 m in ArcMap 10.7 using the bilinear method. Backscatter contains a lot of noise and is only beneficial for estimating vegetation height when it is aggregated [36,39]. The 245m resolution was selected to be comparable to the previous study using the backscatter, InSAR, and Fusion techniques performed in central Maine, U.S., which utilized a pixel that covered 6 ha [39]. The LiDAR data were split using a random mask made in ArcMap 10.7 using Create Random Raster. In total, 70% of the pixels were used for training while 30% were used for testing. The RLCMS was then used to remove all non-forest areas from the training and testing datasets. Backscatter layers were made corresponding to the LiDAR training and testing datasets. The relationship between vegetation height and backscatter power was modeled using Equation (4) [33,39], simplified from [36]:

$$\gamma_0 = A(1 - e^{-Bh_v^C}). \tag{4}$$

A, B, and C are fitting coefficients. $h_v$ is the vegetation height. When fitting the model, $h_v$ is the known vegetation height from the LiDAR training dataset. When the model is used for estimation, $h_v$ is the output FSH. $\gamma_0$ is the backscatter power from the annual mosaic. Simple, empirical relationships can often provide results comparable to those produced by more complex equations [26].

A manual iterative process was employed to fit the A, B, and C coefficients in Python. A total of 20 sets of coefficient values (A, B, and C) were trained and tested using the training LiDAR data and corresponding backscatter power data (Appendix C). Although the values of the coefficients can vary a lot over different test sites, starting values were used from the Maine use case: A = 0.11, B = 0.0622, and C = 1.014 [26,33,39] as well as determined by using the Python Scipy v1.4.1's Curve_fit package. This package utilized non-linear least squares to fit Equation (4) to the LiDAR and backscatter training datasets and suggest appropriate values for A, B, and C [62]. The best backscatter results were obtained with the values A = 0.63152915, B = 0.01037093, and C = 0.9223795. Coefficients generated for different aggregation levels were also explored and, while pixels with areas less than 6 ha did show a decrease in $R^2$ and RMSE, larger aggregations did not show improved metrics due to inhomogeneity, thus only the results for these coefficients at the 6 ha pixel size are shown.

### 2.2.2. Interferometric SAR (InSAR) Technique

The InSAR technique (Figure 4) takes advantage of the height dependence of random motion in the apparent InSAR correlation, i.e., the taller the forest, the larger the wind-induced temporal decorrelation [26,33]. Coherence has been shown to have advantages over backscatter for estimating FSH above 10 m [39]. This is based on temporal decorrelation, as the changes over time between two scenes causes decorrelation in the phase between two images, in addition to the temporal decorrelation induced by dielectric property change [33,63]. Temporal decorrelation has been shown to increase as vegetation height increases [33,64].

The ALOS L1 scenes were processed using ISCE [57] v2.4.1 [58] into interferograms for input into the FSH algorithm v1.2 [39,65]. The spatial resolution of the output interferogram was 30 m. The windows used for range and azimuth multi-looking were 1 and 5, respectively, as assumed by the FSH software [65]. The window of correlation estimation is two times the range/azimuth multi-look averaging window, and this is $2 \times 10$ in this case.

Options to implement rubbersheeting were selected to mitigate ionospheric impacts on the interferogram.

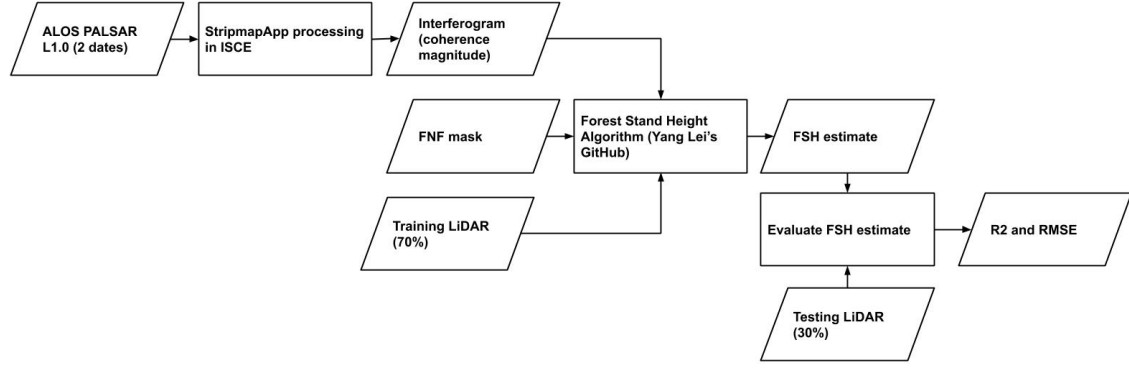

**Figure 4.** Workflow for estimating forest stand height with ALOS PALSAR L-band data using an InSAR-based method.

Coherence, $\gamma$, is the amount of similarity in phase between two passes of a SAR sensor over the same location and quantifies the accuracy of the InSAR phase measurements. $\gamma$ can range from 0 to 1, where 0 is completely decorrelated and 1 is completely correlated [27]. Coherence is defined by $E_1$ and $E_2$, the signals that the sensor received on pass 1 and pass 2, respectively (Equation (5)) [27,39,51]:

$$\gamma_0 = \frac{(E_1 E_2^*)}{\sqrt{\langle |E_1|^2 \cdot |E_2|^2 \rangle}}$$ (5)

$\gamma$ has three components (Equation (6)):

$$\gamma = \gamma_{geo} \cdot \gamma_{SNR} \cdot \gamma_{v\&t}$$ (6)

$\gamma_{geo}$ is the decorrelation caused by viewing geometry, or look angle difference, between the two observations. $\gamma_{SNR}$ is the thermal noise inherent to the sensor. $\gamma_{v\&t}$ is the coupled decorrelation caused by scattering due to vegetation and changes in the observed surface between the two passes attributed to the passes being separated by time.

The relationship between interferometric correlation and in situ vegetation height can be modeled using an inverse sinc function (Equation (7)) [33]. $\gamma_{v\&t}^{HV}$ is the coherence from an HV polarized channel of the SAR sensor. $h_v$ is the in situ data, which in this case are the ground truth proxy LiDAR data. $C_{scene}$ represents the random motion of volume scatterers while $S_{scene}$ represents dielectric property changes that can be related to changes in soil moisture. The upper bound of $h_v$ is $\pi {}^* C_{scene}$.

$$|\gamma_{v\&t}^{HV}| = S_{scene} \operatorname{sinc}(\frac{h_v}{C_{scene}})$$ (7)

This equation assumes that weather and other impacts on $C_{scene}$ and $S_{scene}$ are consistent across the entire scene, which may not be the case [39,51]. Please see Appendix A for a brief investigation into precipitation over the study area.

The quality of the six ALOS L1 scenes available was examined using the mean coherence value (Table 2) calculated in ArcMap 10.7 for overall scene, the forest class, and the forest class over Dongsithuane PF, as determined by the forest/non-forest mask. The Dongsithuane PF corresponds to the LiDAR data available for the study area. Scenes from 25 April 2008 and 8 June 2007 were not included as there were no scenes within an adequate time frame to make an interferogram pair. Coherence values below 0.2 indicated poor data quality for the purposes of the InSAR technique. Ideally the coherence values should be 0.4 or above within forested areas and consistent across the study area [66,67]. Acquisition dates with a small temporal baseline are preferred. There is an inverse relationship between

the length of the temporal baseline and the strength of the relationship between coherence and vegetation height.

**Table 2.** Mean coherence for interferograms available over the study area.

| Date Pairs | Overall | Forest Class | Forest Class in the Dongsithuane PF | Temporal Baseline |
|---|---|---|---|---|
| 8 September 2007 and 9 December 2007 | 0.16 | 0.14 | 0.35 | 92 days |
| 13 June 2009 and 29 July 2009 | 0.15 | 0.20 | 0.33 | 46 days |
| 16 June 2010 and 16 September 2010 | 0.15 | 0.15 | 0.35 | 92 days |

Ultimately, the 2009 interferogram was selected as it was acquired closest to the proxy groundtruth data temporally and had slightly better quality indicated by the mean coherence value over the forest class in the Dongsithuane Production Forest. However, consistency across the image is also key to interferogram quality, which is not addressed in Table 2. In addition, the two interferograms produced with data from 2007 and 2010 did not produce improved FSH estimates when briefly explored. For this technique, the scenes selected should avoid severe weather conditions for both dates [33]. This use case serves as an example of these methods in a challenging scenario, as both scenes were acquired during the wet season.

### 2.2.3. Fusion Technique

The FSH algorithm relies on the fusion of the InSAR technique, in order to improve backscatter-based estimates [26,33,39] (see Figure 5). The FSH estimates from both methods were combined in ArcGIS 10.7. Where the backscatter-based estimate was 10 m or greater, the InSAR-based estimate was used. Fusing at a threshold of 10 m was found useful in the previous use case in the northern-boreal forest in central Maine [33,39], and so that threshold was used as a starting point for this analysis.

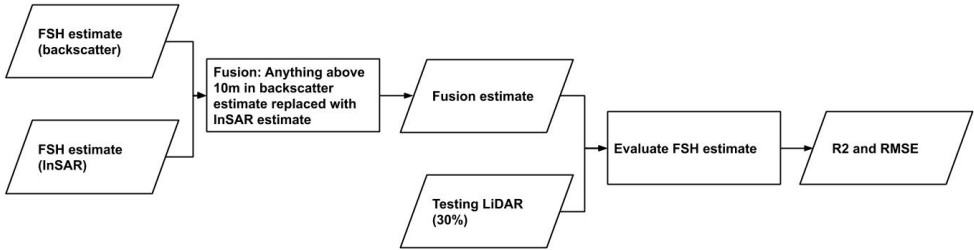

**Figure 5.** Workflow for fusion of the backscatter and InSAR estimation methods.

### 2.2.4. Comparisons

The three estimated height maps, based on the backscatter, InSAR, and fusion methods, were validated with the simple, randomly stratified 30% split of the LiDAR data resampled to 245m. These three height estimate maps were also compared with height estimates from GEDI L2 [52] and height estimates from GLAD [22]. The GEDI data were left in point form and the corresponding pixel values were evaluated. The GLAD data were resampled to 245m from the original resolution (see Table 1). To create a baseline, the agreement between the comparison products was also investigated. For this comparison, the GEDI data were again evaluated against the corresponding pixel value for LiDAR and GLAD datasets in native resolution. The LiDAR product was resampled using the bilinear method to 25 m to compare with the LiDAR dataset to create a reference point for expected results based on an optical technique for this study area and period. The metrics used were the coefficient of determination, $R^2$, as calculated using Scipy v1.4.1's Linregress function in Python, RMSE, as calculated using Scikit-learn v0.22.2's Metrics package, bias, and standard deviation in Python.

An exploration was also conducted into using Random Forest (RF) in GEE to estimate FSH. Three tests were completed with the following inputs: (1) backscatter and interferogram, (2) Landsat 7 bands 1–3 and 5–7, and (3) both SAR and optical inputs. Please see Appendix D. This exploration found the backscatter power from the ALOS annual mosaic to have the greatest importance to the RF model.

## 3. Results

Estimated FSH maps for each method and the distribution of pixels at each height interval are shown in Figure 6. Inset maps of the difference between the LiDAR FSH and the FSH estimated by each method highlight the differences in the estimations. The backscatter, InSAR, and fusion-based FSH estimates are compared in a kernel density plot with the GEDI, GLAD, and LiDAR-based heights as well as through the error metrics: $R^2$, RMSE, and bias (Figure 7). Heights above 13 m, the category including the greatest area for all three estimation methods, are distributed spatially in approximately the same locations, although a pixel-by-pixel inspection reveals that the methods do not produce exactly the same results. The distribution of pixels by height is very different for between the backscatter and InSAR methods.

The median of the backscatter-based FSH estimate is equal to 11. The backscatter-based FSH sample distribution, with a kurtosis value close to zero and skewness value above one, is not too peaked but is negatively skewed. The backscatter-based estimate has a greater maximum than the InSAR-based estimate: 49 m. This tall height may indicate greater uncertainty in the estimate as the backscatter-based techniques can saturate for higher tree heights, increasing uncertainty for higher trees. The backscatter-based estimate compared with the testing LiDAR had an $R^2$ equal to 0.26 and an RMSE equal to 4.90 m (Figure 7).

The median of the InSAR-based FSH estimate is equal to 11. The distribution of the InSAR-based FSH estimate is very negatively skewed and peaked, due to the large amount of area identified as having a height of 14 m (Figure 6). This saturation is likely due to low $C_{scene}$ values, which occur when the InSAR data quality is not optimal. For this estimation, the area within 14 m is over three times larger than the second largest height category, 10 m Figure 6. This cap on estimated height at 14 m can also be seen in Figure 7, where InSAR-based heights are compared with testing LiDAR, GLAD, and GEDI heights. The InSAR-based estimate compared with the testing LiDAR had an $R^2$ equal to 0.19 and an RMSE equal to 3.46 m (Figure 7).

The fusion map of the backscatter and InSAR-based estimates also had a median equal to 11. Due to the 10 m threshold, the fusion-based estimate is very negatively skewed and peaked (Figure 6). This threshold caused the large area of forest identified as 14 m in height from the InSAR-based method to be included in the fusion estimate. Fine-tuning the threshold for this forest type was not pursued, as the InSAR-based heights saturated before the backscatter-based heights in this study area.

In addition to being compared with the LiDAR testing data, the backscatter, InSAR, and fusion-based estimates were compared with GEDI and GLAD products (Figure 7). The backscatter-based estimate performed better in $R^2$ when compared to airborne LiDAR and GEDI products ($R^2 = 0.26$ for both) than when compared with the GLAD Global Canopy Cover product ($R^2 = 0.13$). The InSAR-based estimate performed better in $R^2$ when compared to airborne LiDAR and GLAD products ($R^2 = 0.19$ for both) than when compared with the GEDI L2 product ($R^2 = 0.02$). The fusion map performed best when compared with GLAD ($R^2 = 0.19$), similarly well when compared with LiDAR ($R^2 = 0.18$), and poorly when compared to GEDI ($R^2 = 0.04$). FSH maps from the comparison products along with basic statistics can be found in Figure A3. The LiDAR had a median height of 9 m, while the GEDI had a median height equal to 11 m, and the GLAD a median height equal to 4 m. While the GLAD identifies many areas of lower forest height, many of these areas correspond with non-forest areas. The GLAD and GEDI products were also compared with the LiDAR in order to provide a baseline of performance with established products, these

results can be seen in Figure A2. It should be noted that GLAD reports the Global Canopy Cover metrics at a 30 m resolution with an $R^2$ of 0.61 and an RMSE of 9.07 m compared with their validation LiDAR [22].

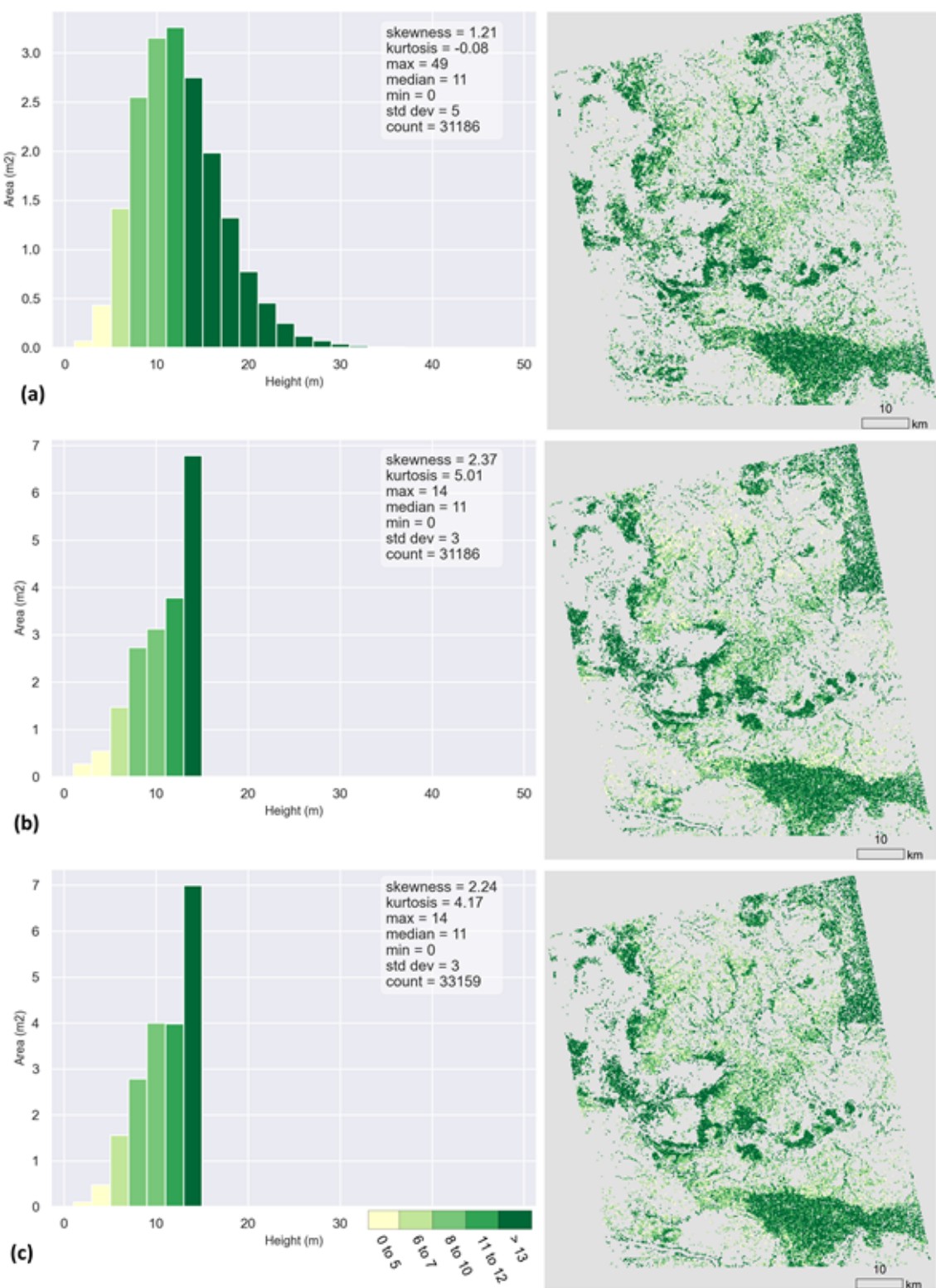

**Figure 6.** The estimated FSH maps and distributions over the study area produced by each method: (**a**) backscatter, (**b**) InSAR, and (**c**) fusion. The difference between the estimated heights from each method and the proxy ground truth height in the LiDAR area is included in the inset.

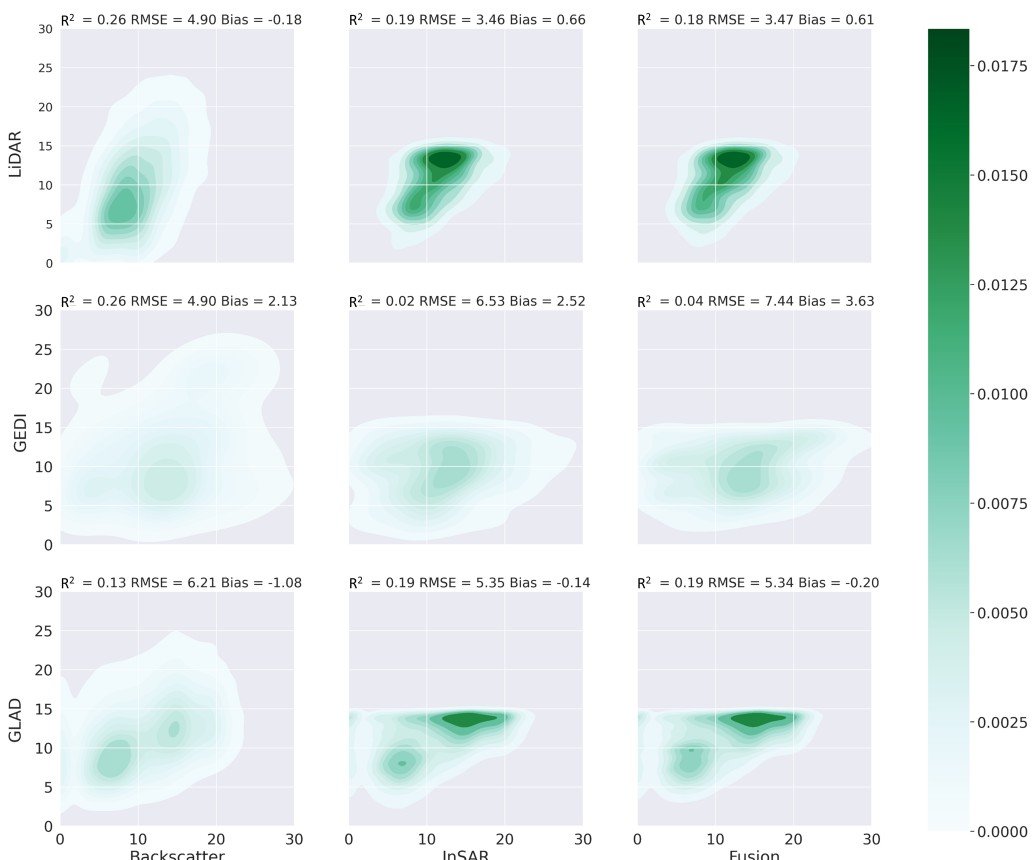

**Figure 7.** Density plots comparing LiDAR height with backscatter-based, InSAR-based, and fusion-based height estimates.

## 4. Discussion

Although the R² and RMSE were improved for the backscatter-based method as compared to the InSAR-based and fusion methods, the maximum heights were not well represented. The backscatter method was slightly more successful in predicting taller forest stands. The distribution of backscatter, InSAR, and fusion height estimates over the Dongsithuane PF, in addition to other comparison products (Figure 6), illustrate to what extent each dataset represents the actual spread of tree heights at that location. These distributions provide an indication of the performance of the estimation methods in capturing the distribution of the ground truth heights. The similarity between the distributions over the entire study area and the area covered by ground truth data indicates that if a product underestimates or overestimates over the LiDAR area, it may do the same across the whole study area. As LiDAR was the proxy ground truth data, the more closely a method captures the spread of the LiDAR height estimates, the more confidence in the results of that method. This comparison of the spread indicates that InSAR and fusion techniques do not capture the taller forest stands. A brief investigation into linear equations relating backscatter to FSH did not yield improved results (Appendix C).

Although the InSAR method was intended to compliment the backscatter method, as backscatter-based estimates are not accurate above 10 m in height, in this case the InSAR-based, and thus fusion-based, estimate actually saturated at a lower height than the backscatter-based method due to the poor data quality and low availability of InSAR at this site. GLAD and GEDI-based heights also have a range more similar to the LiDAR than the InSAR and fusion-based methods, with a maximum of 30 m estimates. Thus, while the InSAR method may add value to FSH estimation, in this use case the tallest quartile of forest stands are not captured well. In addition, the negative skew of the InSAR-based estimates indicate that the InSAR data quality is low.

Based on the error metrics $R^2$ and RMSE when compared with the proxy groundtruth LiDAR data, the InSAR method performed better in this use case. However, these statistics do not tell the whole story. Based on the density plots and the metrics of the backscatter and InSAR-based height estimates when compared with other products, the backscatter performs better overall. Compared with LiDAR, the fusion method had an improved RMSE and only a negligible decrease in $R^2$. Between the two use cases, the $R^2$ tended to perform better for the use case in central Maine, U.S. [33,39,51], while the RMSE tended to perform better for the Lao PDR use case when compared with LiDAR data. For backscatter, the $R^2$ performed worse and the RMSE performed better for Lao PDR compared to the central Maine use case, with an $R^2$ of 0.2 and 0.4, respectively, and an RMSE of 4.9 and 6.12 m, respectively [39]. For InSAR, the $R^2$ performed comparably and the RMSE performed better for the Lao PDR use case compared to the Maine use case, with an $R^2$ of 0.19 and 0.17. The fused $R^2$ performed worse for the Lao PDR use case, presumably because the backscatter $R^2$ performed poorly. The fused RMSE performed better for the Lao PDR use case than the Maine use case. The low $R^2$ and high RMSE for the Lao PDR use case in comparison to the central Maine, U.S. use case could be due to a lack of dynamic range in tree heights in the Lao PDR training dataset. In other words, the majority of the training data was available for trees less than 9 m in height, thus the InSAR method may not have been an appropriate choice for capturing taller tree stands.

### 4.1. Limitations

The main limitation associated with the Lao PDR use case is availability and quality of the ALOS PALSAR data. First, there were only three scene pairs available covering the study area in Savannakhet, Lao PDR. This means that low coherence interferograms could not be avoided. Coherence above 0.2, which is consistent across the scene, is recommended for the InSAR approach [66,67]. For example, the selected pair from 2009 had a mean coherence of 0.33 over the forest class in the Dongsithuane Production Forest and a mean coherence of 0.15 overall. Many areas of random phase were located in forested areas where it is desirable to estimate forest stand height. The decorrelated phase in these areas can contribute to poor forest height estimates [39]. Diurnal and seasonal trends have an impact on the moisture fluctuation and thus the dielectric properties of the surface [68]. All three available image pairs had one or both scenes falling within the wet season, which may have introduced dielectric changes that caused coherence to be low. To mitigate the impacts of seasonal changes, both scenes selected for the final analysis were within the same season. Similarly, the times that these two scenes were taken were within one minute of each other, approximately 15:38 and 15:37 for the center of the scene, and so the diurnal impacts should be minimized. However, investigating a study area with both dates in the dry season would make the conclusions more robust.

Another limitation is the availability and characteristics of the LiDAR data. The LiDAR heights were only available over the Dongsithuane Production Forest, and not distributed across the entire study area, which could cause uncertainties when applying the InSAR and backscatter models created using the training LiDAR data across the larger study area. In addition, with a median equal to 9 m, the majority of the Dongsithuane Production Forest is less than 10 m in height. Thus, InSAR method may not be optimally tuned for this location. However, while the backscatter method is more appropriate for estimating FSH below 10 m, the quality of the SAR data available for this study period may be negatively impacted by the weather causing greater uncertainty.

Topography causes geometric distortions in SAR data, such as foreshortening and layover, and can impact the radiometric measurement of backscatter [27,69]. The greater topographic variety along the eastern edge of the study area, which raises into foothills along the Annamite Mountain Range, may negatively impact the results. However, the area covered with proxy ground truth data where the estimation model is created, and evaluated, should not be impacted as this is a low lying area.

Finally, the backscatter method did not perform as well as previous work utilizing JAXA's ALOS PALSAR annual mosaic for estimating vegetation height [36,39]. This could be due to the limited quality of the input data. However, the outputs may be improved through additional iterations of manual adjustment of the coefficients for Equation (4). Equation (4) is a simplification of an equation developed for AGB estimation by [36]. Further investigation into this more complex equation could improve height estimation based the backscatter products for this study area.

*4.2. Future Work*

This use case defines a pathway forward for applying this method to tropical forests, informed by the limitations and challenges encountered in the Savannakhet, Lao PDR use case. Although objectively low, the error metrics of this method are reasonable as compared to other studies described in Section 1.2. In addition, statistics such as RMSE will appear to perform better for global study areas compared to provincial study areas as included in this work. The results suggest that, with the future launch of NASA-ISRO SAR (NISAR) and its shorter revisit time and greater availability of recent data, the limitations associated with the data are expected to be reduced and thus the benefits of this method will be more readily realized. The workflow outlined in this and the previous case study in central Maine [26,33] can be used as a springboard for integrating L-band data from the future NASA-ISRO SAR (NISAR) mission into remote sensing methods utilized for forest monitoring and REDD+ reporting efforts.

As part of the Second Biomass Retrieval Inter-comparison eXercise (BRIX-2) being conducted in advance of the NISAR launch, our team is leading an effort to implement improvements to the FSH algorithm. These improvements include incorporating more up to date, and denser time series of data, by using ALOS 2. More data may eliminate poor performance issues in two ways, first by allowing for poor quality data to be discarded, and second by allowing a time series of estimates to be utilized. In addition, with the shorter temporal baseline of ALOS 2, the impact of weather on dielectric properties between pairs may decrease. In addition, a pair with both images acquired within the dry season, when there is assumed to be less moisture fluctuation, may be available. Another improvement is the integration of GEDI as a training dataset, which will make this algorithm more feasible for testing globally. The $C_{scene}$ and $S_{scene}$ parameters from the inversion model (Equation (7) are assumed to be constant across the entire image; however, the precipitation patterns are quite different from west to east across the province. Lei et al. (2019) intend to segment the scene to allow for more accurate estimations [39]. The improvements will make an effort towards reaching the pixel size of less than 1 ha that the BRIX-2 activity requires. These suggested improvements will also make this method more applicable for REDD+ reporting. For example, the Intergovernmental Panel on Climate Change (IPCC) good practice recommendations include having a time series of data [12]. While the IPCC does not proscribe specific levels of precision, accuracy and levels of bias [12], quantifying the uncertainty of this method and resulting AGB estimates is imperative considering the potential implementers of this method.

As additional L-band data becomes available, the performance of SAR-based methods can be investigated in additional study areas. These use cases could explore the capacity of the FSH algorithm in regions with even greater topographic variety, such as the Himalayas, various forest types, etc. Additionally, use cases with optimal input data should be examined, for example focusing on forest stands with average heights above 10 m, regions with a greater availability of higher quality SAR data, and incorporating the algorithm improvements discussed above.

**Author Contributions:** Conceptualization, H.B.P., T.M., A.I.F.-A. and K.E.H.; methodology, H.B.P., T.M., K.E.H. and Y.L.; software, Y.L., H.B.P. and T.K.; validation, H.B.P.; formal analysis, H.P.; investigation, H.B.P.; data curation, H.B.P.; writing—original draft preparation, H.B.P.; writing—review and editing, T.M., K.E.H., Y.L., R.G., A.I.F.-A., Q.N. and T.K.; visualization, H.B.P. All authors have read and agreed to the published version of the manuscript.

**Funding:** This research was funded by the joint U.S. Agency for International Development (USAID) and National Aeronautics and Space Administration (NASA) initiative SERVIR and particularly through the NASA Applied Sciences Capacity Building Program, NASA Cooperative Agreement NNM11AA01A.

**Acknowledgments:** The authors would like to thank Rajesh Thapa, SERVIR-Mekong and SilvaCarbon for support in training and data acquisition.

**Conflicts of Interest:** The authors declare no conflict of interest. The funders had no role in the design of the study; in the collection, analyses, or interpretation of data; in the writing of the manuscript, or in the decision to publish the results.

## Abbreviations

The following abbreviations are used in this manuscript:

| | |
|---|---|
| AGB | Above-Ground Biomass |
| ALOS | Advanced Land Observing Satellite |
| ALS | Airborne Laser Scanning |
| ASF | Alaska Satellite Facility |
| CHIRPS | Climate Hazards Group InfraRed Precipitation with Stations |
| DEM | Digital Elevation Model |
| FREL | Forest Reference Emission Levels |
| FSH | Forest Stand Height |
| GEDI | Global Ecosystem Dynamics Investigation |
| GLAD | Global Land Analysis and Discovery |
| GEE | Google Earth Engine |
| IPCC | Intergovernmental Panel on Climate Change |
| InSAR | Interferometric SAR |
| ISCE | Interferometric SAR Computing Environment |
| JAXA | Japan Aerospace Exploration Agency |
| LiDAR | Light Detection And Ranging |
| MRV | Monitoring, reporting, and verification |
| NISAR | NASA-ISRO Synthetic Aperture Radar |
| PDR | People's Democratic Republic |
| PALSAR | Phased Array type L-band Synthetic Aperture Radar |
| REDD | Reducing Emissions from Deforestation and Forest Degradation |
| RF | Random Forest |
| RLCMS | Regional Land Cover Monitoring System |
| RMSE | Root Mean Square Error |
| SAR | Synthetic Aperture Radar |
| SRTM | Shuttle Radar Topography Mission |

## Appendix A. Precipitation Investigation

To obtain a sense of the precipitation that may have impacted the quality of the interferogram used for this analysis, Climate Hazards Group InfraRed Precipitation with Station data were used [60]. These data were acquired at 0.05-degree resolution for 11–13 June and 27–29 July 2009, following [33]. Precipitation measurements were in mm/day. The dates of both scenes utilized in the study fall within the monsoon season. Statistics, including mean, maximum, minimum, and standard deviation of the accumulation of precipitation, were calculated for two days prior to as well as during the day of acquisition of the ALOS L1 data (Table A1). Theoretically, there will be a moisture-induced dielectric impact on the phase for the July 29th scene as compared to the June 13th scene due to the variability in precipitation expected during a monsoon study period.

Change in the dielectric property involves factors beyond weather, as soil type also has a large impact on how moisture is held [70]. Small changes in soil moisture content, even if only a few percentage points, can make large changes in the dielectric constant. In turn these will make large phase differences as shown in interferograms [71]. Thus, even though there may not have been precipitation on June 13, it would be necessary to examine the

days before as well to see if the soil could be saturated from previous precipitation events. Furthermore, there is some uncertainty from using the CHIRPS dataset as it incorporates gauge and satellite data to estimate precipitation in an area. This means that in areas that do not have a lot of available gauge data there may be more uncertainty. In addition, the gauge data comes with its own uncertainties depending on methods used for water collection. Perhaps most importantly, these data for July 29th show that the moisture has greater variation across the scene. This suggests that the dielectric constant is also not evenly distributed across the scene.

**Table A1.** Accumulated precipitation from CHIRPS for the area of the fusion-based FSH estimate.

|                    | 11–13 June 2009 | 27–29 July 2009 |
| :---: | :---: | :---: |
| Mean | 5.8 | 36.5 mm/day |
| Maximum | 13.0 | 23.0 mm/day |
| Minimum | 0 | 57.5 mm/day |
| Standard Deviation | 2.5 | 6.2 mm/day |

**Appendix B. Fitting Coefficients for Backscatter Approach**

A manual iterative process was employed to fit the A, B, and C coefficients in Google Colaboratory. This script can be found on GitHub: https://github.com/HBaldwin3 /CaseStudy_FSH_LaoPDR, accessed on 6 September 2021.

**Appendix C. Alternative Backscatter Approach**

Two alternatives were briefly explored for the backscatter approach. First, the ASF RTC was examined for use instead of the ALOS PALSAR mosaic. The ASF RTC product is in $\gamma_0$ [72]. Based on this small exploration, it is preferable to use the JAXA ALOS PALSAR annual mosaic over the ASF RTC product. Based on the validation and comparison efforts, the ASF RTC product tended to underestimate tree height. $R^2$ and RMSE were calculated in Google Colaboratory for the FSH estimates created using the ASF RTC product and JAXA annual mosaic. This script can be found in GitHub https://github.com/HBaldwin3 /CaseStudy_FSH_LaoPDR, accessed on 6 September 2021.

Second, a linear regression based on the training LiDAR data and corresponding backscatter was used to create a location-specific relationship between backscatter from the annual ALOS PALSAR mosaic and the training LiDAR dataset:

$$\gamma_0 = 0.0033(h_v + 0.0182) \tag{A1}$$

where $\gamma_0$ was the backscatter from ALOS PALSAR and $h_v$ was the estimated FSH. Equation (A1) was then applied to the JAXA ALOS PALSAR mosaic. The more complex equation ultimately used in the full analysis performed better than the linear equation in $R^2$ and RMSE (Table A2).

Finally, the height values from both the backscatter produced FSH estimate maps were compared with the LiDAR testing dataset. $R^2$ and RMSE were calculated for the FSH estimates created using the ASF RTC product and JAXA annual mosaic (Table A2). Ultimately, the JAXA annual mosaic using Equation (4) [33] was used to incorporate in the fusion product. The more complex equation based on [33] performed slightly better than the simple linear regression approach. However, the time required to iteratively fit the coefficients for this equation may negate the small increase in accuracy.

**Table A2.** Error metrics for FSH estimated with backscatter, comparing approaches and products.

| Backscatter Product | Approach | $R^2$ | RMSE (m) | Bias (m) | Compared to |
| :---: | :---: | :---: | :---: | :---: | :---: |
| Annual mosaic | [33] | 0.26 | 4.9 | 0.18 | testing LiDAR |
| ASF RTC | [33] | 0.12 | 6.3 | −1.84 | testing LiDAR |
| Annual mosaic | Linear | 0.23 | 7.05 | 0.42 | testing LiDAR |

## Appendix D. Random Forest Comparison

Three models were trained in GEE. The first utilized only Landsat 7 data. The second utilized the same backscatter and interferogram products used for this study. The third utilized both. The same LiDAR training data were used for all of the models. These models were not tested with the LiDAR or other data. However, the importance of each variable was tested. For both approaches including SAR data, the backscatter power from the annual ALOS PALSAR mosaic was found to have the greatest importance. The importance of the model utilizing all inputs is shown in Figure A1.

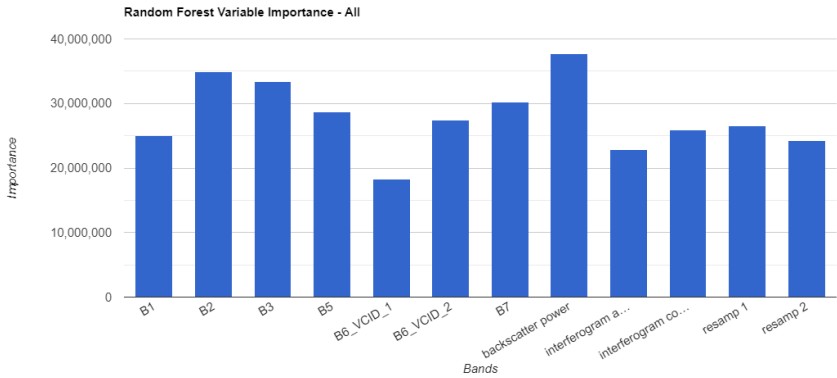

**Figure A1.** The importance for each variable included in the random forest investigation including both optical and SAR inputs.

## Appendix E. Comparison Products

$R^2$, RMSE, and bias were calculated for the height estimates based on the InSAR, backscatter, and fused methods compared to the portion of LiDAR reserved for testing (Figure 7). In addition, the height estimation maps were compared with the GEDI L2 canopy and GLAD height products. Finally, the GLAD [22] product was compared to the LiDAR and GEDI L2 available in the study area (Figure A2). The aim of this comparison was to provide a baseline of performance considering the limited training and validation data in the area, including the limited amount of GEDI points.

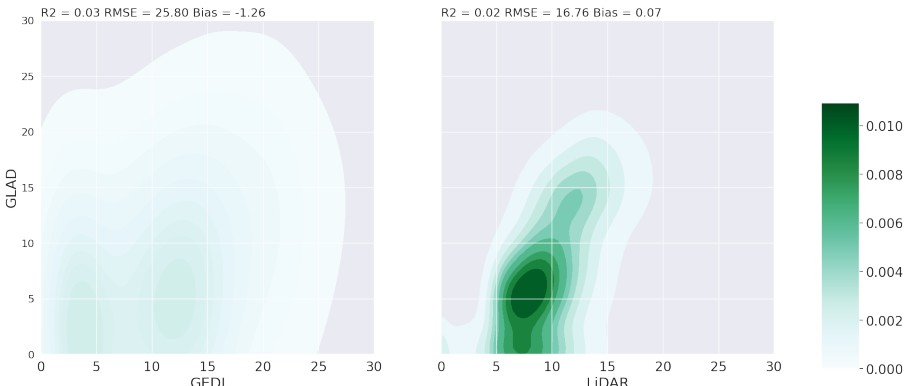

**Figure A2.** The estimated FSH maps and distributions over the study area for the comparison data products: LiDAR and GEDI L2.

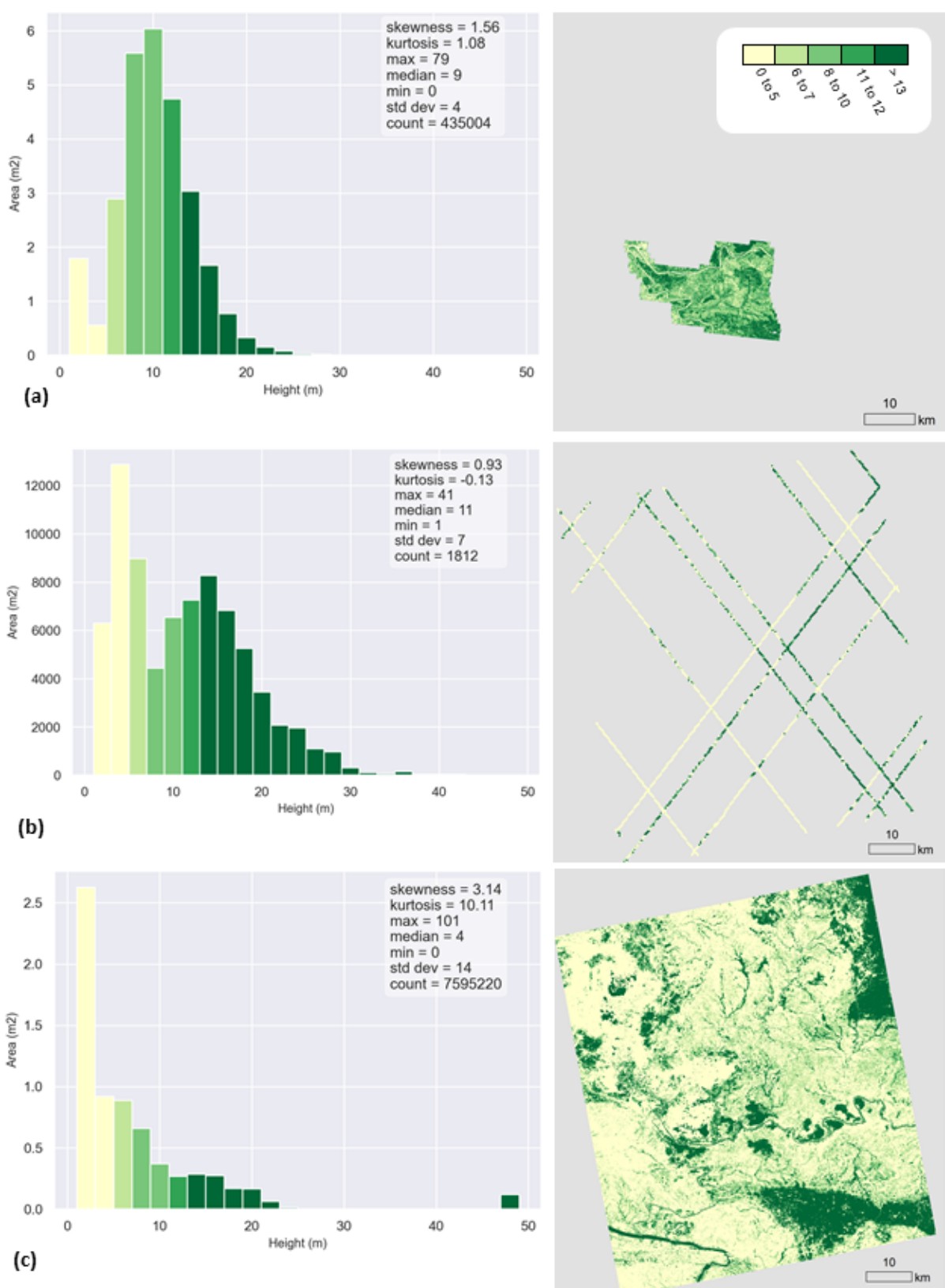

**Figure A3.** The estimated FSH maps and distributions over the study area for the comparison data products: (**a**) LiDAR, (**b**) GEDI L2, and (**c**) GLAD.

## Appendix F. Scripts and Data

All scripts and data used for this case study can be found through the GitHub page: https://github.com/HBaldwin3/CaseStudy_FSH_LaoPDR, accessed on 6 September 2021.

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
