# Peer review of "Estimating Forest Stand Height in Savannakhet, Lao PDR Using InSAR and Backscatter Methods with L-Band SAR Data"

_remotesensing, doi:10.3390/rs13224516_

Round 1

Reviewer 1 Report

The authors present a sort of review of methods based on the fusion of different kinds of data, considering SAR backscatter and InSAR coherence. The paper is generally well written. As for other papers that I've recently seen, the authors must justify better the significance of funding different information than analyzing separately the different information in a comprehensive way. The introduction is sufficiently informative on the state-of-the art of the used technologies. Maybe, something more related to the use of polarimetric SAR data could be added. The results are adequate even not well articulated. The same for the discussion section.

Reviewer 2 Report

General comments:

This research case estimated forest stand height in the study area using InSAR and backscatter methods with L-band SAR data and compare the results with GEDI and GLAD products. Although it is an interesting topic, I can’t find the scientific significance or practical applications of the paper for the reason that neither new methods were proposed nor valuable results for the study area. The authors directly adopted the algorithm and the fusion threshold of 10m from the reference, which is not be appropriate for the study area due to the different forest types and environmental characteristics. The general framework is not reasonable using poor quality and different cover of data for modelling, and comparing the results with the different temporal products from different covers. Generally, the methodology is short of novelty and the results are of little significance.

Specific comments:

  1. Line 7-8: If compared with LiDAR, both RMSE and R2 should be the increment, and the unit of RMSE was missed.
  2. Line 64-70: several researches are listed and the accuracy results were given, but what are the estimated parameters? And how about the unit of RMSE?
  3. Line 90-93: Do evergreen broad leaf and mixed forest not belong to forest?
  4. Figure 3-5: R2 should be R2.
  5. Line 219: 25m2 should be 25m.
  6. Line 233-234: Only20 sets of values were used as trained and tested data. It seems that the samples are insufficient for non-linear model.
  7. Line 298-302: The results of comparisons with GEDI L2 and GLAD can not verify the advantage or validation of the fusion method due to the very different cover areas and data acquisition time, especially using the area of different FSH as the indicator (figure A3).
  8. Line 316-322: what does ‘median of 11’ mean?
  9. Line 321-322: The median height value can not indicate the differences or similarity among GEDI, GLAD and LiDAR due to the different covers and data acquisition times.
  10. Line 362-365: The results can not demonstrate the validity of the fusion method due to the poor data quality and availability of InSAR at the site. Such being the case, what is the significance of this research?
  11. Appendix C: the unit of RMSE and Bias was missed in table A2.

  12. Appendix E, Line 520: what is the table?

Reviewer 3 Report

This paper investigates the possibility to use two forest height estimation algorithms for an enhanced height estimation. The presented experiments are technically sound, however, the manuscript must be improved with additional clarifications to increase its scientific relevance and consistency.

1) Please define clearly which height is addressed, i.e. is it a top height or mean height?

2) line 45-47: having claimed this objective, which conclusion can be drawn? can the mentioned recommendations be supported?

3) The review at lines 73-78 is questionable. Interferometric methods are by far the most established ones for estimating forest height. The use of backscattered power is only limited to few experiments. 

4) Section 1.3: there is no indication of the range of heights to be estimated, and the related suitability of the chosen methods.

5) It is many times claimed that the backscatter-based height saturate at 10 m, however this value is chosen as threshold. On which rationale? And why heights in Fig. 6(a) go well beyond this value? Please clarify / be consistent.

6) lines 199-201: why using soil moisture to explain backscatter changes when speaking about forest heights? Please provide a better discussion.

7) line 255-257: what are this processing options? add explanations if it is necessary to report them in the text. 

8) Section 2.2.4: How meaningful is the comparison with the GEDI footprint-based heights given the strong resolution difference? please comment.

9) Figure 6: the three height map look the same, please provide a better representation to highlight differences / estimation errors. 

10) In general,  my feeling after reading this paper is that the potentials and limitations of the presented method(s) are not adequately addressed. Indeed, the presented experiments only report about the limitations, but there is no discussion about the potentials. Is there a chance to actually show an improvement brought by the fusion? please address.

Round 2

Reviewer 2 Report

Even though the authors added some explanations to my comments, the key issues are not answered and revised. More critically, the study area and data used can not illustrate the validity of the method or are insufficient to support the conclusions. So I still think it necessary to add some data or study area and extend the experiments to prove the robust of the conclusions.

Author Response

Thank you for your review. The authors agree that another study area may have data more suitable for this methodology (in fact, this was demonstrated in the previous use case in central Maine, US, over a northern-boreal forest (Lei, et al., 2019)). The objective of this study is to apply these methods and evaluate in a real world use case relevant to potential practitioners, rather than present a new approach or suggest that this approach is ideal (Section 1.4 Objectives, lines 115-117: “Previously used to estimate FSH for the state of Maine, US [34–36], this use case demonstrates the applicability of the FSH estimation method in a tropical region with limited data availability.”). Our justification for this type of applied research paper is that practitioners need to be informed of the qualities and limitations of this approach as they evaluate their available data and various FSH estimation methods. In order to provide value to these practitioners, the authors included an exploration of available SAR data for an area of interest to partners and an assessment of its quality (lines 279 - 296, including Table 2) and limitations of the approach (Section 4.1 Limitations), including an investigation into potential weather impacts (Appendix A). While the authors had originally hoped to provide a concrete assessment of this method for tropical forest vs. boreal forest, this was not possible due to the limitations outlined (Section 4.1) and was excluded from the conclusions. In regards to your concern about the nature of the conclusions, text has been added to lines 453-456 in Section 4.2 Future Work to clarify the intent of the paper and the extent to which conclusions can be drawn. This now reads, “The results suggest that, with the future launch of NASA-ISRO SAR (NISAR) and its shorter revisit time and greater availability of recent data, the limitations associated with the data are expected to be reduced and thus the benefits of this method will be more readily realized.” The overarching conclusion stated in lines 440-442 was reorganized to focus on the need for taking into account the limitations outlined in the paper, it now reads: “This use case defines a pathway forward for applying this method to tropical forests, informed by the limitations and challenges encountered in the Savannakhet, Lao PDR use case.” Through this reorganization, the authors aim to clarify that no conclusion can be made about this method being more appropriate for tropical forests than another method with the data available at this location. However, the limitations outlined provide guidance for the potential application in other scenarios.

Reviewer 3 Report

I still believe that the review of the InSAR methods for estimating forest height is poor and partially misleading. There is indeed a large variety of algorithms for estimating forest height from InSAR data with or without polarimetry able to yield better results than the one presented by the authors.  There is a wide literature on this, including airborne and spaceborne results. The fact that an algorithms based on temporal decorrelation is used is just dictated by one of the (long-term) objectives of this research, i.e. to motivate the suitability of NISAR data. Completing this literature review is definitely required to recommend publication.

Author Response

Thank you for your review. The authors agree that there are numerous methods for estimating forest stand height using InSAR. The intent of Section 1.2 was not to suggest that this InSAR/Backscatter method is the best performing method, but rather why this method has come about and emphasize that it is relevant for decision makers to view the outputs of this method as applied to a case study beyond central Maine, US (Lei et al., 2019). The background provided has been restructured to clarify how this method fits into previous work. Specifically, in Section 1.2: Estimating Forest Stand Height with Remote Sensing, the paragraph describing how remote sensing is relevant to estimating FSH (and hence, relevant to REDD+) in lines 54-55 now leads into the use of optical data with examples, including the recent UMD product (lines 55-67). SAR-based approaches have been moved to a separate paragraph. How SAR provides additional value is shown in lines 68-71 and then provides examples of other SAR-based methods in lines 72-76. The authors understand that the reviewer would find the inclusion of more examples on InSAR methods and what has been done in tropical forest to establish the history of these methods are provided in lines 77 -87 and available (although limited) recent InSAR as used for estimating FSH in tropical forest (lines 91-93). Although a comparison of InSAR estimation methods is not within scope, This outline provides context for why this method is being investigated in additional study areas, as the objectives of this paper are to apply these specific methods in a real world context and assess the limitations in order to communicate these to potential practitioners (Section 1.4).